# Investigating diversity in European audiences for public engagement with research: Who attends European Researchers' Night in Ireland, the UK and Malta?

Aaron Michael Jensen[1,2], Eric Allen Jensen[1,2,3]*, Edward Duca[4], Joseph Roche[5]

1 Qualia Analytics, Dublin, Ireland, 2 Institute for Methods Innovation, Arcata, California, United States of America, 3 University of Warwick, Coventry, United Kingdom, 4 University of Malta, Msida, Malta, 5 Trinity College Dublin, Dublin, Ireland

* eric@methodsinnovation.org

**Data Availability Statement:** All relevant data are uploaded to the Zenodo repository and publicly

## Abstract

European Researchers' Night is an annual pan-European synchronized event devoted to public engagement with research. It was first held in 2005 and now occurs in over 400 cities across Europe, with the aim of bringing researchers closer to the general public. To investigate social inclusion in these events, we conducted survey research across three national contexts (Ireland, Malta and the UK) and events in seven cities between 2016 and 2019 (n = 1590). The results from this exploratory descriptive study confirmed one hypothesis, namely that event attendees had substantially higher levels of university qualification than the national publics. This is in line with wider patterns of unequal participation in public engagement with research activities based on socio-economic status. However, we also found mixed evidence on the prevalence of ethnic minority representation among event attendees compared to the general population, thus failing to uphold the second hypothesis that predicted an over-representation of white majority participants. This second finding diverges from existing research findings about ethnic diversity amongst science communication audiences, raising the possibility that some public engagement events are over-performing on this dimension of social inclusion. Overall, the findings demonstrate that European Researchers' Night has potential for addressing the critical goal of enhancing the diversity of audiences for public engagement with research, even as it falls short on the key metric of socio-economic diversity.

## Introduction

Like many governments around the world, the European Union (EU) has been focused for many years on improving engagement between the scientific community and European society. A pressing concern is the question of who from within European society has the opportunity to engage with, participate in and benefit from research and innovation (e.g., [1]). In an effort to widen public access and engagement, a pan-European initiative called European

accessible at https://zenodo.org/record/4905718#. YMdn0fkzY2w.

**Funding:** This study was supported by the European Union's Horizon 2020 Research and Innovation Programme (https://ec.europa.eu/ programmes/horizon2020) in the form of a grant awarded to ED as part of the Science in the City project (818730), a grant awarded to JR as part of the PROBE project (817914), a grant awarded to JR as part of the QUEST project (824634), and a grant used to fund part of the work of AJ and EJ on this paper as part of the Cork Discover project (818789). The funders had no role in study design, data collection and analysis, decision to publish, or preparation of the manuscript.

**Competing interests:** Co-author Edward Duca is the current organizer of a European Researchers' Night event. This could be perceived as a competing interest. However, this does not alter our adherence to PLOS ONE policies on sharing data and materials.

Researchers' Night has taken place across Europe since 2005. Here, we present an exploratory descriptive analysis of respondent characteristics from a sample of those attending European Researchers' Night over a multiple year period across three different countries.

European Researchers' Night (ERN) is a unique event in each country, but as a program of activities, it has similarities with science festivals. However, an important difference is that it promotes the full range of research disciplines, not only science and technology (which is not always the case with science festivals). Because the research literature on science festivals is more developed than European Researchers' Night per se, we focus here on what is already known about science festivals as an approach to engaging public audiences with research.

Historical records indicate that science festivals have taken place in Europe since at least 1831 [2]. The last few decades has seen a dramatic increase in the number of science festivals, as well as their diversity and scale [3]. Bultitude et al. [4] analyzed 94 science festivals from around the world to clarify commonalities. Bultitude et al. define science festivals as celebrations (time-limited and recurring) of scientific ideas and content with the intention of engaging non-specialists. Indeed, this definition also applies to European Researchers' Night. The goal of these events is to make research visible to society, with implicit aims of fostering the idea that EU research is both "transparent and participative" [5]. The event also aims to boost the visibility of research careers, helping to reinforce the 'pipeline' into research careers. Moreover, events like European Researchers' Night are seen by the funder as part of a broader effort to develop more socially responsible research and innovation [6], in line with wider global initiatives to develop socially sustainable research systems that enable meaningful interaction between science and society (e.g., [7]).

Jensen and Buckley [8] investigated the reasons why people attend science festivals and found that "visitors value the opportunities afforded by the science festival to interact with scientific researchers" while the "development of increased interest in and curiosity about new areas of scientific knowledge within a socially stimulating and enjoyable setting" was the most significant self-reported benefit of participation (p. 557). Kennedy et al. [9] investigated science festival audience profile patterns across the UK. They found that 'those attending science festivals do so because they are already interested in and comfortable with science, and tend to be privileged on a number of socioeconomic dimensions' such as educational attainment [9] Specifically, Kennedy et al. [9] report:

> A large proportion of visitors to each festival held bachelor or postgraduate degrees (80% and 45% respectively in the eastern city, 71% and 30% in the southern, and 74% and 31% in the northern). This compares to an average of 38% of UK adults with undergraduate degrees [. . .] and 11% [. . .] with postgraduate degrees.

On the basis of the findings of limited social inclusion in prior research about science festivals, we prioritized investigating whether events under the auspices of European Researchers' Night are likewise skewed towards those with higher levels of educational attainment.

## European Researchers' Night: Background and context

European Researchers' Night is a program that has been running for more than a decade consisting of hundreds of simultaneous public engagement events taking place annually in over 400 cities across Europe [10]. Events in 2019 reported a combined attendance of over 1.6 million people [10]. This program receives European Commission funding, as well as support from government agencies in charge of higher education and culture, and industry sponsors.

The overall aim of the program, as defined in the European Commission's Horizon 2020 Work Programme, is to increase awareness of "research and innovation activities, with a view

to supporting the public recognition of researchers, creating an understanding of the impact of researchers' work on citizen's daily life, and encouraging young people to embark on scientific careers" [11]. In other words, the program seeks to bring researchers and European citizens closer together and promote the sharing of ideas and research in a variety of ways that may be engaging or entertaining. The range of science- and art-themed activities at European Researchers' Night may include live demonstrations, hands-on experiments, performances and workshops. More specifically, activities may include short stories, games, guided tours of research labs, talks, citizen debates, panel discussions, activities, interactive demonstrations and many other forms of engagement.

The European Union has invested tens of millions of euros in European Researchers' Night since 2005. Thus, it is worth considering the available evidence about what this investment has delivered. However, despite a requirement for 'impact assessment' by funded projects, there is a lack of published empirical research on European Researchers' Night in peer-reviewed academic journals. In this paper, we present findings from a large sample frame (N = 2092) of event participants gathered across four years (2016–2019) through on-site evaluations in three countries (Ireland, United Kingdom & Malta).

An evaluation study of the 2015 event in Dublin compared perceptions of European research among the attendees of European Researchers' Night in Ireland and local publics [12]. However, the attendees of the event were not representative of the general public and were more likely to have a third-level (degree level and above) education. Despite a significant investment in marketing and promotion, awareness of the event was low even in public locations close to the university campus where the event was held. This evaluation indicated that European Researchers' Night has strong potential for effective public engagement, albeit with a number of weaknesses in terms of its implementation. These weaknesses were examined in a subsequent consideration of the overall objectives of the event and it was deemed that "for European Researchers' Night to successfully achieve its goal of raising awareness of European research, it needs to have the same level of scrutiny and rigor applied to it as the research it promotes" [13]. This is in keeping with the wider calls for 'evidence-based science communication' [1] and improved evaluation efforts focusing on impact [14].

These engagement events are often organized by museums, universities and science centers. To investigate whether the European Researchers' Night events are attracting a diverse and broadly representative sample of the public, we empirically examined three national contexts. This understanding of the profile of event attendees, we refer to these festivals using national labels. Each of these events is well established and runs annually. All of the events examined rely heavily on volunteer staff, including local university students, staff of universities or museums and members of the general public.

Here, we briefly introduce the events below:

**European Researchers' Night in Malta (2019).** In Malta, European Researchers' Night is the largest national science and arts festival, which attracts an estimated 30,000 people annually in recent years. Reported as one of the largest European Researchers' Nights in Europe, the festival events take place in the streets and buildings of the historic capital city Valletta, which opens widely to provide visitors with access. The range of activities include interactive performances (music, theater and dance, stand-up comedy), art exhibitions (or installations) and hands-on activities (games or experiments). A comprehensive, national marketing campaign reaches around 300,000 people on all forms of media.

**European Researchers' Night in Ireland (2017–2019).** In Ireland, European Researchers' Night includes two different cities (e.g., with data collected for two years running). The Ireland events primarily took place on university campuses, and to a lesser extent in city centers. Activities took place in lecture theaters, labs and other campus spaces that are usually occupied by

academic staff and students. Walk-up events allowed people to take part in tours, discussions, viewing posters and science-themed arts and demonstrations. In the Dublin event, for example, researchers were involved from all three faculties: the faculty of engineering, mathematics, and science; the faculty of arts, humanities, and social sciences; and faculty of health sciences. Audiences for the event were publicly invited through advertising on legacy media (national television and radio) and social media (Facebook) in weeks leading up to the night, as well as posters around the host cities.

**European Researchers' Night in the United Kingdom (2016–2017).** The UK sample for this study encompasses data collected over two consecutive years across four different cities. These events, used to engage new audiences with scientific research projects, took place at university campuses, libraries and science centers. A separate program of events and interactive exhibitions was made available for each location.

**Research aims and hypotheses.** In this paper, we investigate audience profiles and respondent characteristics from a sample of those attending European Researchers' Night events. Demographic details of European Researchers' Night attendees are scant, especially when it comes to audience ethnicity. However, the general trend seen across public engagement events in non-formal learning spaces is an over-representation of white majority participants [15–17]. This has led to calls for change in how such events are run [18, 19] and the need for a more comprehensive understanding of audience profiles. While many of ERN events do not publicly report specific information about the profile of their audiences, even when they publish the evaluation reports required by the European Commission (e.g., [20]), organizers of the ERN in Rome reported key demographic characteristics based on the 2017 iteration of their longstanding event. They reported the following profile results: 'The majority of attendees' had a High School degree (28%) and University degree (Bachelor's degree, 18%, and Master's Degree, 27%). [. . .] The good level of qualification, higher than the Italian distribution [. . .] suggests that the attendees were not casual visitors, but people informed, motivated and interested in knowing more about science' [21].

These indicative figures about the European Researchers' Night audience profile in one site combined with existing evidence from the literature on science engagement audiences (e.g., on minority ethnic participation [22]) was used to formulate the following hypothesis to be explored in the present study:

*Hypothesis 1 (H1)*. ERN events attract an audience with *higher educational qualification levels* than the general population.

*Hypothesis 2 (H2)*. ERN events attract an audience that *over-represents white majority participants* compared to the general population (and under-represent minority ethnicities).

To evaluate whether ERN events are attracting a diverse and broadly representative sample of the public, we empirically examined three national contexts.

Considering the lack of published research on European Researchers' Night, we wanted to investigate whether the educational level of people attending these activities in several locations was representative of the wider public. Attracting diverse audiences is a key aim of European Researchers' Night [10] towards the democratization of research for all European residents. We wanted to analyze whether the lack of diversity seen in science festivals [9] is present in European Researchers' Night events, favoring audiences with a higher level of educational qualification.

## Methods

Secondary data were analyzed for this study. Primary electronic written consent for the anonymized data to be used for research or evaluation purposes (including academic publication)

was gained from all participating survey respondents by the events directly. Therefore, an institutional review board application was not submitted for this work. This section describes the methods and procedures used to gather audience survey responses, the sample frame distribution and the approach to analyzing responses from the achieved sample. The approach employed here represents a balance between the practical comprises needed for real-world naturalistic exploratory research and ideal sampling practices such as ensuring equal probability of selection and random allocation to treatment and control groups. The latter are rarely feasible in audience research settings, where the public has free choice about where they go to spend their leisure time [23, 24].

## Instrument

This questionnaire used closed-ended multiple choice questions (e.g. demographic data and Likert scales about attitudes towards research) and limited open-ended questions (e.g. 'what comes to mind when you think of research'). The questions that were analyzed for this paper focus primarily on quantitative nominal and ordinal data (e.g. household income and educational attainment) in order to compare to wider population data.

The research used a software solution designed for paired samples with matching between pre-visit and post-visit responses at the individual level, as well as automated email invitations and reminders for the post-visit questionnaire and real-time data analysis and automatic visualizations for event organizers.

## Procedure

The survey instrument was administered in English following the same set of protocols across all the participating sites for this study. Data collection occurred in two steps: On the day of the event, adult attendees were approached by data collection volunteers and asked if they were willing to provide answers to a few questions on site, and then respond to a follow-up survey sent by email after the event. Respondents who gave consent were enrolled (pre-visit survey) by providing email accounts and later received an initial invitation to participate in the post-visit survey and up to two reminders.

Pre-visit questionnaires were administered to visitors in each city by data collectors using a systematic on-site 'intercept' sampling approach, as is standard in audience research settings to mimic random selection to the extent feasible (e.g. Jensen & Lister, 2015). These data collectors served as research assistants and received training prior to the start of each event. Data collectors used tablets to gather responses from people attending walk-up (no booking required) events. Where demographic questions were potentially sensitive (e.g. regarding ethnicity or income), respondents were either given the tablets to complete their answers independently or these questions were saved for the post-visit survey. All sites used software for offline and GDPR-compliant data collection provided by the research technology company Qualia Analytics (qualiaanalytics.org).

## Data analysis

Our presentation of findings in the report uses unweighted data. That is, no adjustments have been made to reflect the probability of particular respondents being selected and completed the pre-visit survey on-site. As presented in this report, we have been careful in how far we extend claims from the responses that have been provided. Furthermore, instances where total percentages add to less than 100 are due to either rounding of decimals, exclusion of response categories (i.e., "unsure/don't know") or questions that have multiple response options (e.g.,

'tick all that apply'). The primary software tool for the analysis was Qualia Analytics' built-in dashboard, supplemented by limited use of Microsoft Excel for data management.

National population statistics were used as a basis for comparison to address the hypotheses in this study [25–29] In Malta, the National Statistics Office did not, at the time of this study, identify ethnicity in its national (census) survey, therefore ethnic diversity could not be analyzed with the Maltese sample. This prevented Maltese data from contributing to assessment of Hypothesis 2.

## Sampling

We analyzed audience profiles for those attending of European Researchers' Nights held in 7 cities across the UK, Ireland and Malta between 2016 and 2019. This analysis produced descriptive statistics about the audience profiles of European Researchers' Night events across each country. For all events, the total sample frame of invited respondents (N = 2092) was dispersed between Malta (24%, n = 498), Ireland (37%, n = 779) and UK (39%, n = 815). The response rate comparison for attendees by country is presented in **Table 1**.

The sample frame distribution by year of data collection is presented in **Table 2**.

## Results

This study was designed to assess audience profiles in terms of demographic diversity and representativeness of the wider public. Here, we begin with levels of educational qualification as a key indicator of social inclusion.

### Hypothesis 1: Educational attainment

Most pre-visit respondents indicated having at least some university-level education (68%, n = 994), with most holding degrees at undergraduate (33%, n = 481) or postgraduate (35%, n = 513) levels (**Table 3**).

We found that a large proportion of adult attendees to each European Researchers' Night event were more highly educated than the national populations (See **Table 4**). Indeed, a large portion of attendees held undergraduate (Malta: +26%; Ireland: +17%; UK: +17%) or postgraduate degrees (Malta: +24%; Ireland: +33%; UK: +23%) than the respective national populations. Compared with national figures for each country, these figures have shown an overrepresentation of university educated attendees (or degree holders). For example, as a combined segment, degree holders attending ERN events in Malta and Ireland was +50% greater than the respective national populations, while UK events were +40% greater. By comparison, ERN attendees with qualifications below university degree (Malta: -1%; Ireland: -21%; UK: -14%) or no qualification (Malta: -48%; Ireland: -8%; UK: -23%) were much less prevalent in the sample compared to the respective national populations.

**Table 1. Sample frame distribution of ERN survey respondents by country.**

| Country | n = | % |
|---|---|---|
| Malta | 498 | 24 |
| Ireland | 779 | 37 |
| UK | 815 | 39 |
| Total | **2092** | **100** |

The data collection was conducted across multiple years. The total sample frame of respondents was distributed between 2016 (24%, n = 492), 2017 (26%, n = 548), 2018 (16%, n = 330) and 2019 (35%, n = 722).

**Table 2. Sample frame distribution by year of data collection.**

| Year | n = | % |
|---|---|---|
| 2016 | 492 | 24 |
| 2017 | 548 | 26 |
| 2018 | 330 | 16 |
| 2019 | 722 | 35 |
| Total | **2092** | **100** |

The survey was carried out on the day of each event through face-to-face intercept data collection conducted at entrances to the events. For all events, the total sample frame of invited respondents was split between those willing to participate (83%, n = 1590; achieved sample size) and those who declined to participate (17%, n = 317) at the point of intercept. Specific levels of achieved sample size are indicated in the results based on available data for each analysis.

More specifically, ERN attendees in the sample across all years and countries have shown underrepresentation of those with 'no qualification' compared with national population statistics. While a disparity was evident in all countries, the extent of this disparity was most evident in Malta, where almost half (48%) of its population is reported in national statistics as having no formal education qualification. At the same time, in contrast to other countries, ERN attendees in Malta with 'below university qualifications' were similar to the national statistics.

## Hypothesis 2: Ethnic diversity

A majority of respondents were White (86%, n = 618), with the second most prevalent self-identified ethnicity being Asian (9%, n = 64), followed by African, Caribbean or Black (2%, n = 17) and Mixed or multiple ethnic groups (3%, n = 18) (**Table 5**).

We found that the ethnic diversity of the adult attendees of the European Researchers' Night events were similar to or greater than the respective national populations (See **Table 6**). For example, a slightly greater portion of attendees in the UK (+2.3%) were reported as White, but those identifying as White in Ireland were less prevalent (-15.6%) than the national population. Likewise, the proportion of UK ERN attendees who self-identified as Asian was similar to the national population (-0.1%), but Asian participants were more prevalent in Ireland ERN events (+13%) than the population. By comparison, African, Caribbean or Black participants were slightly less prevalent in the UK events when compared to national official statistics

**Table 3. Comparison of educational attainment of respondents across all event locations.**

| | n = | % |
|---|---|---|
| No formal education qualification | 2 | 0 |
| Primary education | 47 | 3 |
| Secondary education | 287 | 20 |
| Vocational qualification | 118 | 8 |
| First University Degree (Bachelor's or equivalent) | 481 | 33 |
| Postgraduate Degree (Master's, PhD or equivalent) | 513 | 35 |
| Total | **1448** | **100** |

Those respondents with university education identified a range of subjects, including Science (n = 136), Humanities (n = 106), Social Science (n = 87), Technology (n = 76), Biological Science (n = 41), Mathematics (n = 19), and Literature (n = 21), Engineering (n = 9) and Health (n = 13). Many respondents chose 'Other' (27%, n = 191), as the list of degrees was not comprehensive.

**Table 4. Comparison of educational attainment of national population and participating ERN audiences.**

| | | National Population | ERN Samples | |
|---|---|---|---|---|
| Qualifications | Country | % | % | ± % |
| No qualification | Malta | 48 | 0 | -48 |
| | Ireland | 8 | 0 | -8 |
| | United Kingdom | 23 | 0 | -23 |
| Below Undergraduate Degree | Malta | 42 | 41 | -1 |
| | Ireland | 44 | 23 | -21 |
| | United Kingdom | 47 | 33 | -14 |
| Undergraduate Degree | Malta | 6 | 32 | +26 |
| | Ireland | 18 | 35 | +17 |
| | United Kingdom | 15 | 32 | +17 |
| Postgraduate | Malta | 3 | 27 | +24 |
| Degree | Ireland | 10 | 43 | +33 |
| | United Kingdom | 11 | 34 | +23 |

(-1.5%) but such ethnic minority participants were slightly more prevalent in Ireland ERN events compared to the national population (+2.3%). Event participants from Mixed or multiple ethnic groups were slightly more prevalent in the UK events (+0.3%) and also in Ireland (+1.5%) compared to the respective national population statistics (**Table 6**).

Considering the overall relationship between Ethnicity and Qualification Levels in the sample for this study, we conducted an analysis to assess and found no statistically significant relationship (DF = 2, $X^2$ = 1.87, p = .392).

## Discussion

In this paper, we focus on demographic variables indicative of social inclusion and, where possible, compare the results to demographic population data for the wider populations of all three countries. This research brings together evaluation evidence from three European countries and several cities designed to assess the diversity of the audience for these public engagement events through an exploratory secondary analysis.

Based on the descriptive findings presented here, we show that European Researchers' Night events have some similarities with other public engagement initiatives in struggling to reach beyond the highly educated publics that normally attend these kinds of events. The evidence we have presented has confirmed the first hypothesis—the European Researchers' Night events we studied attract an audience with higher educational qualification levels than the general population. This finding connects to a larger concern that research institutions find it difficult to reach out beyond their enthusiastic fans [9].

The second hypothesis, that 'ERN events attract an audience that over-represents white majority participants compared to the general population (and under-represent minority

**Table 5. Response rate comparison of ethnicity across ERN events.**

| | n = | % |
|---|---|---|
| White | 618 | 86 |
| Asian | 64 | 9 |
| African, Caribbean or Black | 17 | 2 |
| Mixed or multiple ethnic groups | 18 | 3 |
| Total | **717** | **100** |

**Table 6. Comparison of ethnicity of national population and participating science festival visitors.**

| Ethnicity | Country | National Population | ERN Samples | |
|---|---|---|---|---|
| | | % | % | ± % |
| Mixed | Ireland | 1.5 | 3.0 | +1.5 |
| | United Kingdom | 2.2 | 2.5 | +0.3 |
| Black | Ireland | 1.4 | 3.7 | +2.3 |
| | United Kingdom | 3.3 | 1.8 | -1.5 |
| Asian | Ireland | 1.7 | 14.6 | +13 |
| | United Kingdom | 7.5 | 7.4 | -0.1 |
| White | Ireland | 94.3 | 78.7 | -15.6 |
| | United Kingdom | 86.0 | 88.3 | +2.3 |

In Ireland, we compared the ethnicity of the European Researchers' Night attendees to the ethnicity of the student body at the university campus where the events were held in Dublin [30]. The student body profile (White, 91%; Asian, 5%; Black, 2%; Mixed, 2%) was more closely aligned to the national population than to the audience of the European Researchers' night events.

ethnicities)' could not be upheld. Data to address this hypothesis were available from samples in two countries: In Ireland, there was a -15.6% underrepresentation of White participants compared to population, +13% Asian and +2.3 Black. In the UK, there was a small level of over-representation of white majority participants in line with the hypothesis (+2.3% White). Taken together, we have clear evidence of over-representation of ethnic minority groups at engagement events in one country, and a small level of over-representation of White ethnic groups in the other. This mixed bag of evidence for ethnic minority representation in public engagement with research audiences is insufficient to uphold Hypothesis 2.

Social inclusion is an important issue within the public engagement field and is one of the key priorities for the European Union's Cohesion Policy 2014–2020 and the Europe 2020 Strategy [31]. Indeed, there is a longstanding problem of social inequality in public engagement with research, wherein those from more privileged backgrounds are more likely to participate in these types of experiences (e.g., Kennedy et al. 2018). Moreover, there is evidence that public engagement professionals may inadvertently design the content and structure of events in a way that is more appealing for audiences from socio-economically advantaged backgrounds [32].

Overwhelmingly, public engagement activities like European Researchers' Night are serving a highly educated audience, albeit one that is more ethnically diverse than their national populations. But even if ERN organizers were able to attract more educationally diverse audiences, their programs and exhibitions may not be satisfying to these newly diversified audiences. Those who are current non-visitors may not feel like these public engagement opportunities are appropriate for them [33, 34]. The key is for public engagement organizers to establish their relevance and how the event content links to people's daily lives (also [35]).

In conclusion, Kennedy et al.'s [9] argument based on science festival audiences apply equally to public engagement events under the auspices of European Researchers' Night when it comes to educational attainment, which is a well-established indicator of socio-economic status [36]. In principle, research should be for the benefit of everyone, regardless of social class. It is well-established in social research that socio-demographic variables can predict access and progress in formal education. Public engagement events that operate outside of formal educational institutions should ideally be working to reduce such patterns of de facto exclusion. Public engagement should help to make sure that research "not an exclusive club like fine art, opera, or other forms of high culture, [instead it] should ameliorate—rather than

reinforce—disparities in access" [9]. There is clearly a "need for fundamental change. . . to achieve true, socially inclusive 'quality'" public engagement [37]. Along the way to that change, event organizers can find encouragement in the finding of at least some positive evidence for ethnic diversity amongst European Researchers' Night audiences.

## Acknowledgments

The authors would like to acknowledge the support of the volunteers that helped gather data, and the event coordinators for the European Researchers Nights studied for this paper. Simone Cutajar was a particularly important contributor for the evaluation work taking place in Malta.

## Author Contributions

**Conceptualization:** Aaron Michael Jensen, Eric Allen Jensen, Joseph Roche.

**Data curation:** Aaron Michael Jensen.

**Formal analysis:** Eric Allen Jensen.

**Funding acquisition:** Aaron Michael Jensen, Eric Allen Jensen, Edward Duca, Joseph Roche.

**Investigation:** Eric Allen Jensen, Edward Duca.

**Methodology:** Aaron Michael Jensen, Eric Allen Jensen.

**Project administration:** Aaron Michael Jensen, Eric Allen Jensen, Joseph Roche.

**Resources:** Eric Allen Jensen, Edward Duca, Joseph Roche.

**Software:** Aaron Michael Jensen, Eric Allen Jensen.

**Supervision:** Aaron Michael Jensen, Eric Allen Jensen.

**Validation:** Eric Allen Jensen.

**Writing – original draft:** Aaron Michael Jensen, Eric Allen Jensen.

**Writing – review & editing:** Eric Allen Jensen, Edward Duca, Joseph Roche.

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
