## [Decision Letter · Decision Letter 0]

18 Mar 2021

PONE-D-20-39168

Investigating Diversity in Public Engagement with Research: 

Who attends European Researchers’ Night in Ireland, UK and Malta?

PLOS ONE

Dear Dr. Jensen,

Thank you for submitting your manuscript to PLOS ONE. After careful consideration, we feel that it has merit but does not fully meet PLOS ONE’s publication criteria as it currently stands. Therefore, we invite you to submit a revised version of the manuscript that addresses the points raised during the review process. To accept your revised version for publication, I invite you to address and/or respond to all the comments, suggestions, and critics reviewers highlighted for your work.  

We look forward to receiving your revised manuscript.

Kind regards,

Sina Safayi, D.V.M., Ph.D.

Academic Editor

PLOS ONE

Journal Requirements:

"Edward Duca is the current organizer of a European Researchers' Night event. This could be perceived as a competing interest."

Reviewers' comments:

Reviewer's Responses to Questions

**Comments to the Author**

1. Is the manuscript technically sound, and do the data support the conclusions?

Reviewer #1: Partly

Reviewer #2: Yes

Reviewer #3: No

2. Has the statistical analysis been performed appropriately and rigorously? 

Reviewer #1: Yes

Reviewer #2: Yes

Reviewer #3: No

3. Have the authors made all data underlying the findings in their manuscript fully available?

Reviewer #1: Yes

Reviewer #2: Yes

Reviewer #3: Yes

4. Is the manuscript presented in an intelligible fashion and written in standard English?

Reviewer #1: Yes

Reviewer #2: Yes

Reviewer #3: Yes

5. Review Comments to the Author

Reviewer #1: I would suggest some significant revisions related to hypothesis 2.

First, the authors could say more about why they propose hypothesis 2. The previous evidence shared in the paper suggests that audiences for ERN events may be more likely to have higher educational qualifications (hypothesis 1), but there is no literature cited by the authors to explain why the research team thought that ERN events would attract white participants. Citing more scholarship on the lack of racial/ethnic diversity in STEM fields would be helpful to better motivate/present hypothesis 2.

Second, while the data is sound, the data does not support the conclusions regarding hypothesis 2. This is why I rated the paper as only partly supportive of the data. The data supports conclusions regarding hypothesis 1. In the analysis of hypothesis 2, the discussion of ethnic minority participants states that the ethnic diversity of the adult attendees were similar to or greater than their respective populations. However, this was not the case in the UK, so I found this to be a puzzling conclusion given the data shown in table 6. Whites were over-represented by 2.3% in the UK and Blacks were under-represented by 1.5%. In addition, the authors don't share the ethnic data for the Malta event and do not explain why the ethnicity data is not reported for the Malta event. What did the study find about ethnic minority participants in Malta? Based on the incomplete data presented here, it would be more accurate to conclude that there were national differences in ethnic minority participation in ERN events. So results were inconclusive regarding hypothesis 2.

Reviewer #2: This paper studies two hypotheses with regard to attendees at European Researchers' Night in Malta, Ireland and UK:

1. That the attendees are university educated

2. That the attendees are predominantly from the white majority population

I have a number of minor comments:

1. The data in Table 1 is a little confused with regard to secondary education. It seems to include the Irish data for Senior and Junior Certificate and the remaining countries under Secondary Education. I recommend consolidating secondary education under one heading.

2. In the results section, it becomes clear that only adult attendees were surveyed. I recommend stating this point clearly in the Methods section.

3. The statement in the abstract about "over-inclusion" of ethnic minorities in ERN activities is over stated: it depends very strongly on one immigrant community, Asians, in one country, Ireland. The comments in the results section are more measured and accurate. No attempt is made to discuss the significant differences in the demographics of the Asian immigration to Britain and Ireland. It may be that the age and education profile of Asian immigrants in Ireland is very significantly different.

Overall, I find the study makes a valuable contribution to the debate on whether much Science Communication effectively engages the wider public. I recommend to publish after minor revisions.

Reviewer #3: Public engagement with research is an important topic and should be researched for its impact and efficacy. This paper explores who attends European Researchers’ Night (ERN) – a type of public research engagement event series in three different countries as a proxy for impact. The goal is to further research’s reach to broader spectrum of our societies.

The authors find that attendance at ERN events is skewed with more individuals with college degree or higher when comparing national population. There are huge variations in the attendance based on ethnicities represented at these events as compared to their representation across the population of their own countries. The authors claim the education status representation of individuals in attendance was as expected, whereas the ethnic representation was unexpected.

This manuscript, as it stands now, is not a true research study. It is missing a) sound hypotheses, b) comparisons to controls, c) statistical analyses, and d) correlations with and influences of major causal factors.

Here are the main drawbacks of this article-

1. The rationale and evidence behind hypothesis 2 (expectations for attendance distribution between white and non-white demographics) is not provided questioning the purpose of this study.

2. Socioeconomic and education status are conflated with each other and used interchangeably throughout the paper – the authors need to be clear what they are studying and need to articulate the relationship between education and socio-economic status based on evidence.

3. There is no mention of the merits of or the biases introduced via the “intercept” data collection in the demographic representation.

4. The sample size claim is misrepresented. Out of the 2092, 317 declined to participate in their study.

5. The authors fail to address the influence of the event location on the educational status and ethnic representation seen in the events. For example, events in Ireland were hosted on university campuses and thus likely attracted more educated people including immigrant researchers (contributing to more nonwhite people attending). Local demographic comparisons are lacking – for example, population demographics of the city where the event was hosted in comparison to event attendance is a comparison to be made. In a country like Ireland, there likely are huge variations in demographics between urban & rural areas thus influencing their ultimate data.

6. The authors claim that ERN event demographics mirror other science festival events, although the direct comparison of attendance with previous events is not provided. Overall, the data severely lack controls or statistical analysis to make claims. How does this specifically compare to data on other attendance data such as museum memberships, science festival attendances, outreach event attendances? Is it statistically different?

7. The primary source and the year of the national population data used for comparisons are not noted – this is important as the data in the paper are across many years and it is known that national demographics change with time.

8. The correlation of educational status to ethnic representation is not considered or even discussed – are some ethnicities overrepresented in people with or without college degrees?

9. The authors make blanket conclusions about ethnicity data despite huge differences in the demographics of attendance pattern UK vs Ireland. Moreover, ethnicity data from the country of Malta is completely missing. The authors fail to acknowledge this.

Other items-

10. The survey language is not provided in the Methods section.

11. The exact type of activities and specific examples that constitute the ERN events are sparsely described and no information is provided for activity types in the UK whatsoever.

12. The overall writing flow and logic is hard to comprehend and is extremely disjointed.

6. PLOS authors have the option to publish the peer review history of their article (what does this mean?). If published, this will include your full peer review and any attached files.

Reviewer #1: No

Reviewer #2: **Yes: **Brian Cahill

Reviewer #3: No

---

## [Author Response · Author response to Decision Letter 0]

28 Apr 2021

All the reviewers' comments have been gratefully received and carefully reviewed. All comments that required revisions have been actioned and detailed responses confirming this have been provided in the attachment on responses to reviewers' comments.

---

## [Decision Letter · Decision Letter 1]

25 May 2021

Investigating Diversity in European Audiences for Public Engagement with Research: 

Who attends European Researchers’ Night in Ireland, the UK and Malta?

PONE-D-20-39168R1

Dear Dr. Jensen,

We’re pleased to inform you that your manuscript has been judged scientifically suitable for publication and will be formally accepted for publication once it meets all outstanding technical requirements.

Kind regards,

Sina Safayi, D.V.M., Ph.D.

Academic Editor

PLOS ONE

Additional Editor Comments (optional):

Reviewers' comments:

Reviewer's Responses to Questions

**Comments to the Author**

1. If the authors have adequately addressed your comments raised in a previous round of review and you feel that this manuscript is now acceptable for publication, you may indicate that here to bypass the “Comments to the Author” section, enter your conflict of interest statement in the “Confidential to Editor” section, and submit your "Accept" recommendation.

Reviewer #1: All comments have been addressed

Reviewer #2: All comments have been addressed

2. Is the manuscript technically sound, and do the data support the conclusions?

Reviewer #1: Yes

Reviewer #2: Yes

3. Has the statistical analysis been performed appropriately and rigorously? 

Reviewer #1: Yes

Reviewer #2: Yes

4. Have the authors made all data underlying the findings in their manuscript fully available?

Reviewer #1: Yes

Reviewer #2: Yes

5. Is the manuscript presented in an intelligible fashion and written in standard English?

Reviewer #1: Yes

Reviewer #2: Yes

6. Review Comments to the Author

Reviewer #1: All of my previous reviewer comments were addressed in the revision. The authors both explained that Malta data is not available for ethnic/racial demographics, and noted that their results were mixed on hypothesis 2.

Reviewer #2: The authors have addressed all comments from the first review and I recommend the publication of this submission without revision.

7. PLOS authors have the option to publish the peer review history of their article (what does this mean?). If published, this will include your full peer review and any attached files.

Reviewer #1: No

Reviewer #2: No

---

## [Editor Report · Acceptance letter]

18 Jun 2021

PONE-D-20-39168R1 

Investigating Diversity in European Audiences for Public Engagement with Research:
Who attends European Researchers’ Night in Ireland, the UK and Malta? 

Dear Dr. Jensen:

I'm pleased to inform you that your manuscript has been deemed suitable for publication in PLOS ONE. Congratulations! Your manuscript is now with our production department. 

Kind regards, 

on behalf of

Dr. Sina Safayi 

Academic Editor

PLOS ONE